# ImageNet-trained CNNs are not biased towards texture: Revisiting feature reliance through controlled suppression

**Tom Burgert[1,2], Oliver Stoll[1,2], Paolo Rota[3], Begüm Demir[1,2]**
BIFOLD[1], TU Berlin[2], University of Trento[3]
{t.burgert,o.stoll,demir}@tu-berlin.de, paolo.rota@unitn.it

## Abstract

The hypothesis that Convolutional Neural Networks (CNNs) are inherently texture-biased has shaped much of the discourse on feature use in deep learning. We revisit this hypothesis by examining limitations in the cue-conflict experiment by Geirhos et al. To address these limitations, we propose a domain-agnostic framework that quantifies feature reliance through systematic suppression of shape, texture, and color cues, avoiding the confounds of forced-choice conflicts. By evaluating humans and neural networks under controlled suppression conditions, we find that CNNs are not inherently texture-biased but predominantly rely on local shape features. Nonetheless, this reliance can be substantially mitigated through modern training strategies or architectures (ConvNeXt, ViTs). We further extend the analysis across computer vision, medical imaging, and remote sensing, revealing that reliance patterns differ systematically: computer vision models prioritize shape, medical imaging models emphasize color, and remote sensing models exhibit a stronger reliance on texture. Code is available at https://github.com/tomburgert/feature-reliance.

## 1 Introduction

Convolutional neural networks (CNNs) have played a central role in the development of deep learning models for visual recognition [1], [2], [3], [4]. Their success across a range of computer vision (CV) benchmarks has contributed to the perception that they acquire perceptual representations resembling those of humans [5], [6], [7]. However, a growing body of work suggests that CNNs may process visual information in fundamentally different ways [8], [9], [10]. One of the most influential claims in this direction is that CNNs trained on ImageNet are inherently biased towards texture [8], in contrast to humans who predominantly rely on shape cues [11]. This claim, first formalized by Geirhos et al. [8] through their cue-conflict experiment, has since shaped much of the discourse on how to evaluate and interpret the use of features in deep neural networks.

In the cue-conflict experiment, images are synthesized by combining the shape of one object class with the texture of another, using neural style transfer techniques [12]. Models and humans are then presented with these hybrid images, and their predictions are analyzed to infer which visual cues they rely on. The observed divergence, with CNNs favoring texture and humans favoring shape, has become a dominant narrative for understanding human–machine perceptual differences and has inspired a wide range of follow-up studies [13], [14], [15], [16], [17].

Although influential, the cue-conflict experiment is based on assumptions that may limit the generalizability and clarity of its findings. Conceptually, it reduces feature reliance to a binary choice between shape and texture, overlooking other potentially informative cues such as color, and tends to link salience with reliance implicitly. Methodologically, the generated stimuli entangle unintentionally

39th Conference on Neural Information Processing Systems (NeurIPS 2025).

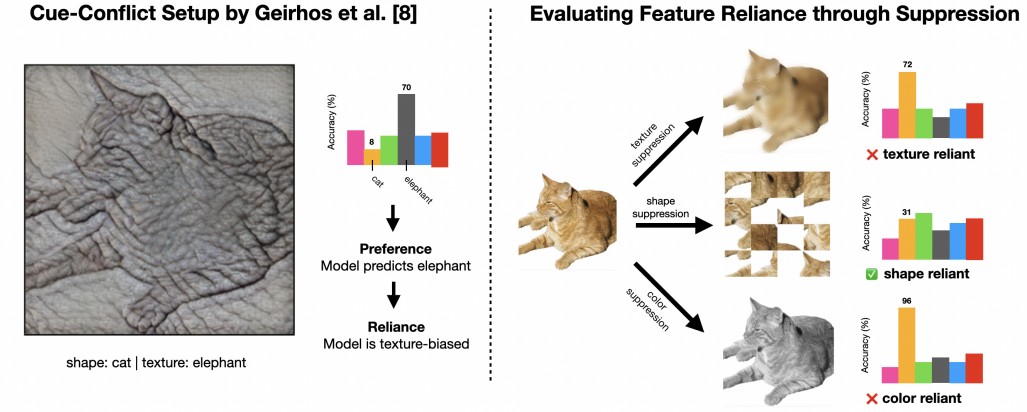

Figure 1: Comparison of cue-conflict setup [8] (left) and our suppression-based framework (right). While Geirhos et al. infer reliance through preference on hybrid images, our framework directly quantifies reliance by measuring accuracy under systematic suppression of texture, shape, or color.

multiple features, introduce texture cues across the image in a spatially unbalanced manner, and rely on shape-based response interfaces that may bias human judgments. As discussed further in Section 3, these conceptual and methodological limitations complicate conclusions about the feature use of models and humans.

In this work, we argue for a conceptual shift: from analyzing feature bias through forced-choice conflicts to assessing feature reliance through targeted suppression. This conceptual distinction reframes how feature preferences and reliance should be evaluated. A model may prefer a certain cue in conflict, not because it is more predictive, but because it is more salient. Conversely, a model may rely heavily on a feature in natural settings, even if it does not dominate in cue-conflict scenarios. To address the aforementioned limitations, we propose a new domain-agnostic evaluation framework that quantifies performance degradation under systematic suppression of individual feature types (e.g., shape, texture, and color), enabling empirical measurement of reliance. The proposed framework does not rely on adversarial inputs or neural style transfer, but instead uses direct feature-suppressing transformations. By isolating individual feature contributions, our framework offers a more reliable basis for interpreting model decisions and comparing representational strategies, both between humans and neural networks, and across model architectures and domains.

Our main contributions are as follows:

(1) We present a re-examination of Geirhos et al.'s cue-conflict experiment [8], highlighting aspects in their evaluation protocol that may limit its generalizability.

(2) We introduce a domain-agnostic framework for evaluating feature reliance through targeted feature suppression, enabling cleaner measurement of model dependence on individual visual cues without requiring conflicting cue setups.

(3) Using the proposed framework, we systematically compare human and model feature reliance under controlled conditions. Our results challenge the texture bias hypothesis [8] by showing that CNNs are not inherently texture-biased; instead, they only exhibit a pronounced sensitivity to local shape, which can be mitigated through modern training strategies. Notably, models trained with vision-language supervision most closely match human behavior.

(4) We apply the same framework to assess domain-specific differences in feature reliance, showing that models trained on CV, remote sensing (RS), and medical imaging (MI) datasets prioritize distinct visual cues depending on domain characteristics.

## 2 Related Work

Understanding which features deep neural networks rely on for image classification has been a long-standing research question. While early interpretations of CNNs assumed a hierarchical buildup from

low-level edges to complex shape representations [5], [6], [7] more recent studies have challenged this view, suggesting that CNNs often rely disproportionately on local texture rather than global shape [9], [10], [8], [18]. Geirhos et al. [8] formalized this observation as the texture bias hypothesis, using a cue-conflict protocol to reveal divergent feature preferences between humans and CNNs.

Subsequent work investigated factors shaping feature reliance beyond architecture. Hermann et al. [13] showed that texture bias in CNNs arises primarily from training objectives and augmentations, with techniques like blurring and cropping increasing shape bias more than architectural changes. Although shape features are present in deeper layers [15], [14], they are not consistently used during classification. Transformer-based models and vision-language models have shifted this discussion. Vision transformers (ViTs) exhibit lower texture bias due to their global attention mechanism [19], [20], and vision-language models show improved alignment with human-like shape use [21].

Various methods have attempted to enforce shape bias or suppress texture cues for improved robustness, including anisotropic filtering [22], edge encoding [23], style disentanglement [24], [25], and shape-focused augmentations [26], [27]. However, stylization alone may improve robustness independent of shape bias [28], and neither shape nor texture bias reliably predicts generalization [16]. These findings have motivated integrative approaches that combine diverse feature biases. Joint supervision [29], ensembles [30], and adaptive recombination [31] aim to harness complementary features. Ge et al. [32] and Jain et al. [17] show that disentangling and combining shape, texture, and color improve robustness and interpretability. Nonetheless, Lucieri et al. [33] caution that in domains like MI, cue entanglement is essential and biasing towards shape may be counterproductive.

Efforts to increase shape bias are often motivated by the broader goal of human-model alignment. Geirhos et al. [34], [35] show that even robust models exhibit error patterns that diverge from humans, revealing a persistent consistency gap. Muttenthaler et al. [36] further argue that alignment with human conceptual structure depends more on training signals than model scale, indicating that robustness and shape bias alone are insufficient proxies for human-like perception.

## 3 Rethinking Texture Bias: A Critical Look at Cue-Conflict Evaluation

The hypothesis that CNNs trained on ImageNet are biased towards texture was popularized by Geirhos et al. [8], who introduced a cue-conflict evaluation protocol. In this protocol, images were generated by neural style transfer [12], combining the shape content (cue) of one class with the texture content (cue) of another. Predictions from both humans and CNNs on these images were then used to infer whether classification decisions were driven more by shape or texture features. Over time, the cue-conflict evaluation protocol has become a de facto standard for assessing feature bias in deep neural networks. While impactful, this protocol introduced several assumptions and limitations that have received limited attention. Conceptually, the protocol frames feature reliance as a binary shape-or-texture choice, which may overlook other cues such as color and conflates preference with dependence. In addition, the stylized stimuli constrain the evaluation of feature bias to naturalistic images with a similar set of classes and cannot be generalized across datasets (e.g., flower classification) or domains (e.g., RS, MI). Beyond these conceptual limitations, the cue-conflict protocol exhibits three methodological concerns in its design and implementation:

(i) **Lack of Feature Isolation.** The texture cues within the cue-conflict images also preserved information beyond texture, including color and local shape structures (e.g., contours and parts of silhouettes). As a result, the synthesized texture cue was not a pure representation of texture but a composite of multiple features, making it difficult to attribute classification behavior to texture alone. An example can be seen in Figure 2a.

(ii) **Overloaded Texture Class Signals.** The protocol consistently inserted texture cues not only into the object region but also into the image background. Since CNNs aggregate local statistics across spatial positions, this broad spatial distribution increases the signal strength of the texture class relative to the shape class. This spatial imbalance systematically biases CNNs towards texture-based decisions, not because of an intrinsic preference but due to the dominant spatial availability of the texture signal. An example can be seen in Figure 2b.

(iii) **Human Interface Bias Towards Shape.** Participants in the human experiments selected the image class by clicking on buttons labeled with icons representing each category. These icons represented global shape characteristics (i.e., silhouettes), potentially guiding participants

towards matching shape features in the cue-conflict image with the icon. This response format potentially introduces bias towards shape decisions, especially when participants were unsure which feature to prioritize. The used icons are visualized in Figure 2c.

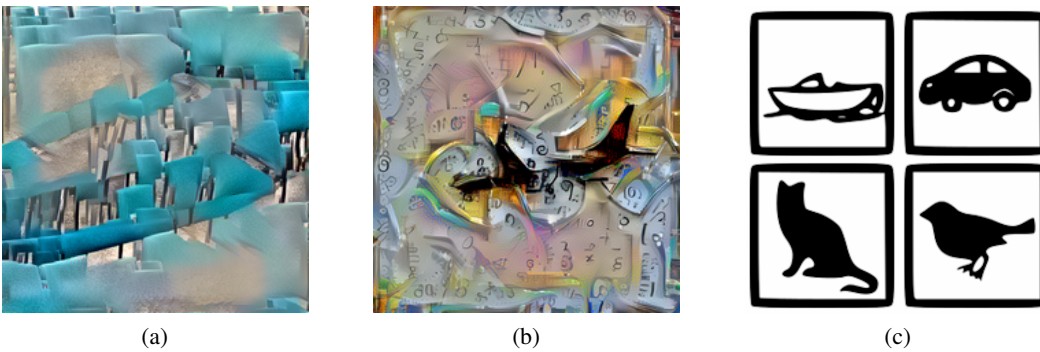

|  (a)  |  (b)  |  (c)  |

Figure 2: Example images taken from the cue-conflict dataset [8]. **(a)** Boat shape cue merged with chair texture cue. **(b)** Airplane shape cue merged with clock texture cue. **(c)** Icons of the human interface to select classes.

These design choices may inadvertently influence CNNs towards texture-driven decisions and humans towards shape-based decisions and complicate drawing definitive conclusions about the actual feature reliance of models.

## 4    A Domain-Agnostic Framework for Feature Reliance

Accurately assessing how deep neural networks rely on different visual features remains a central challenge in understanding their behavior. While the cue-conflict evaluation protocol [8] introduced a reliance test based on feature bias, it imposes conceptual and methodological constraints that limit its generalizability. Rather than forcing models to choose between shape and texture, we propose to assess their reliance on individual feature types by systematically suppressing them and measuring the resulting impact on classification performance. This shift enables a more flexible, generalizable, and semantically grounded analysis of feature use in neural networks.

To evaluate the reliance of deep neural networks on individual visual features, we employ a set of image transformations that selectively suppress shape, texture, or color information while minimally affecting the remaining features. Each transformation is chosen for its ability to target a specific feature class. We define three feature types:

- **Shape** refers to information carried by spatial arrangement and structural contours, including both global (object outline) and local (part-level) shape.

- **Texture** is defined by repetitive patterns, high-frequency local variations, and fine-grained surface details.

- **Color** denotes chromatic information independent of spatial layout or texture.

For each feature type, we include two complementary transformations that differ in their suppression mechanisms and preservation profiles, offering distinct but comparable perspectives on the targeted feature. The transformations are summarized in Table 1 and briefly described in the following. Patch Shuffle [37], [28] and Patch Rotation disrupt shape by modifying non-overlapping image patches: Shuffle randomizes spatial positions, while Rotation preserves locality of patches but breaks edge continuity. Both affect global or local shape, depending on the grid size. Bilateral Filtering [38] and Gaussian Blur reduce texture by smoothing high-frequency details, with the former preserving edges more effectively. Grayscale removes chromatic cues entirely, while Channel Shuffle disrupts color correlations without altering intensity. In the following, we validate the suppression effects of these transformations using quantitative metrics.

Table 1: Feature suppression transformations used in this work. Each feature is suppressed using two transformations with differing strengths.

| Feature Type | Transformation 1 | Transformation 2 |
|---|---|---|
| Shape | Patch Shuffle | Patch Rotation |
| Texture | Bilateral Filter | Gaussian Blur |
| Color | Grayscale | Channel Shuffle |

Table 2: Quantitative validation of suppression transformations across 800 images of ImageNet. Each transformation is used with a fixed parameter setting (see Param ID legend below). Values report normalized metric scores. Arrows indicate desired direction: ↑ higher is better, ↓ lower is better.

| Transformation | Param ID | Texture↓ | Shape↑ | LV↓ | HFE↓ | ESSIM↑ | GC↑ |
|---|---|---|---|---|---|---|---|
| *Texture-Suppressing* | | | | | | | |
| Bilateral Filter | A | 0.521 | 0.796 | 0.548 | 0.493 | 0.737 | 0.855 |
| Box Blur | B | 0.193 | 0.363 | 0.237 | 0.148 | 0.436 | 0.289 |
| Gaussian Blur | C | 0.349 | 0.662 | 0.392 | 0.306 | 0.744 | 0.579 |
| Median Filter | D | 0.357 | 0.506 | 0.399 | 0.316 | 0.584 | 0.429 |
| NLMeans Denoising | E | 0.706 | 0.797 | 0.723 | 0.690 | 0.730 | 0.864 |

| Transformation | Param ID | Texture↑ | Shape↓ | LV↑ | HFE↑ | ESSIM↓ | GC↓ |
|---|---|---|---|---|---|---|---|
| *Shape-Suppressing* | | | | | | | |
| Patch Shuffle | F | 1.000 | 0.176 | 1.000 | 1.000 | 0.205 | 0.147 |
| Patch Rotation | F | 1.000 | 0.293 | 1.000 | 1.000 | 0.339 | 0.247 |

**Legend:** A: $d$=11, $\sigma_c$=170, $\sigma_s$=75; B: $k$=11; C: $k$=11, $\sigma$=2.0; D: $k$=11; E: $h$=20, $tws$=11, $sws$=11; F: grid=6.

## 4.1 Quantitative Validation of Suppression Transformations

While the individual transformations used in this work are not novel, their selection for targeted feature suppression requires empirical justification. To validate that each transformation suppresses the intended visual feature (e.g., texture, shape) while preserving others, we quantify their effects using four metrics: Local Variance (LV) [39] and High-Frequency Energy (HFE) [40] to assess texture suppression, and Edge-SSIM (ESSIM) [41] and Gradient Correlation (GC) to measure shape preservation. All metrics are normalized to the range $[0, 1]$ by dividing by the scores of the unsuppressed (i.e., original) image. Higher values of ESSIM and GC indicate better preservation of edge and structural information, while lower values of LV and HFE reflect stronger suppression of texture features. Further, we compute a harmonic mean across the two texture metrics (Texture) and the two shape metrics (Shape) for each transformation.

We test the effectiveness of the feature suppression transformations across 800 sampled images from the ImageNet validation set. For each transformation, we evaluate a representative parameter setting chosen to balance suppression of the target feature and preservation of others. The respective parameters, such as kernel size or smoothing strength, are indexed by Param IDs in Table 2, with details listed below the table. A full ablation of different parameter settings is provided in the supplemental material (see Section D). In addition to our selected texture suppression transformations, we also compare common alternatives such as Non-Local Means Denoising [42], Box blur, and Median filtering [43] to ensure a fair comparison across standard smoothing techniques. Among texture-suppressing methods, bilateral filtering yields the most balanced trade-off between reducing texture (LV: 0.54, HFE: 0.49) and preserving shape (ESSIM: 0.74, GC: 0.85). Gaussian Blur suppresses texture more uniformly but leads to a greater loss of shape information. Box blur and median filtering remove texture strongly, but at a substantial cost to shape preservation. For shape suppression, we evaluate Patch Shuffle and Patch Rotation with a grid size of 6. These transformations preserve texture but substantially disrupt structural contours, making them suitable for assessing shape reliance. To complement the quantitative evaluation, qualitative visual examples of the suppression effects are provided in the supplemental material (see Section C).

# 5 Experiments

## 5.1 Experiment I: Human vs. CNNs Feature Reliance

**Experimental Setup**. To compare human and model reliance on different visual features, we designed a controlled experiment inspired by Geirhos et al. [34], [8]. We constructed an ImageNet16-like dataset by selecting 50 representative images for each of 16 entry-level categories derived from the WordNet hierarchy [44] (see [34] for details). Images were selected based on the most confidently predicted samples in the ImageNet validation set [45] by a ResNet50 [2] pretrained on ImageNet1k, ensuring balanced subclass coverage. For categories with insufficient confident predictions (airplane, knife, oven), additional samples were manually added. All images were resized to $224 \times 224$ pixels.

Humans were presented with image stimuli in randomized order under one of five conditions: original, global shape suppression, local shape suppression, texture suppression, or color suppression. Each feature was suppressed via a single transformation with fixed hyperparameters: Patch Shuffle with grid size 3 (global shape), grid size 6 (local shape), bilateral filtering with $d$=12, $\sigma_{\text{color}}$=170 and $\sigma_{\text{space}}$=75 (texture), and grayscale conversion (color). See Section 4.1 for justification. Each participant saw only one randomly chosen version of each image to avoid learning effects. The five suppression conditions of one image were split across groups of five participants to ensure balanced coverage. Twenty participants completed the study. Following Geirhos et al.[8], each trial included a $300\,\text{ms}$ fixation square, $200\,\text{ms}$ image presentation, and $200\,\text{ms}$ pink noise mask (1/f spectral shape) to minimize feedback processing. Participants selected one of 16 categories via a $4 \times 4$ grid of alphabetically sorted class names. An additional "not clear" button was available for unrecognizable stimuli. Attention checks were administered every 100 trials, and failed trials were excluded. Additional details and interface screenshots can be found in the supplemental material.

Model evaluation mirrored the human protocol, evaluating their performance under the same five suppression conditions using the identical image set shown to humans. For each image, the class prediction was computed by summing softmax outputs over all ImageNet subclasses mapping to the same entry-level category. Only predictions above the threshold of 0.5 were considered correct. This procedure was chosen heuristically, complementary results using argmax to define class predictions are reported in the supplemental material and show nearly identical reliance profiles.

We evaluated several architectures: ResNet50-standard, trained from scratch with basic augmentations, and ResNet50-sota, trained with a modern recipe [46]. Additional CNNs include MobileNetV3 [47], EfficientNet [3], EfficientNetV2 [48], ConvMixer [49], ConvNeXt [4], and ConvNeXtV2 [50]. Transformer-based models include ViT [51], DeiT [52], SwinTransformer [53], and CLIP ViT [54]. All models except ResNet50-standard were obtained as pretrained checkpoints from the `timm` library [55]. The detailed training procedures can be found in the supplemental material.

**Results.** Figure 3 presents a comparative overview of the performance of humans and CNNs under feature suppression, plotted as the relative accuracy (i.e., accuracy under suppression divided by baseline accuracy on original images). Separate subplots show results for each suppressed feature type. We highlight three representative CNNs: ResNet50-standard, ResNet50-sota, and ConvNeXtV2 alongside human performance. The results show that CNNs are not strongly reliant on texture: under texture suppression, ResNet50-standard retains 80% of its original performance, close to performance under global shape suppression (83%). The highest vulnerability is observed under local shape suppression, where accuracy drops to just 28%. Humans exhibit a similar reliance profile with local shape suppression being most disruptive, but show higher robustness to it (76% retained accuracy). Interestingly, modern training strategies substantially mitigate this effect: the ResNet50-sota reaches 62% under local shape suppression, and ConvNeXtV2 improves further to 65%. These results suggest that the heavy reliance on local shape observed in earlier CNNs is not architectural in nature but can be alleviated through better training regimes. A likely contributing factor is the inclusion of stronger regularization, improved data augmentations, and more extensive training schedules in the modern setup, which may encourage broader feature utilization beyond local patterns. Statistical significance tests confirming these differences are reported in the supplemental material.

Broadening the analysis to a wider range of architectures (Table 3), we observe that several models trained with state-of-the-art recipes exhibit a more balanced reliance profile. However, this trend is not universal: ConvMixer, EfficientNet, and MobileNet variants retain a strong dependence on local shape, indicating that improved training alone does not guarantee human-like feature use and that architectural inductive biases or capacity limitations may still play a role. Among transformer-

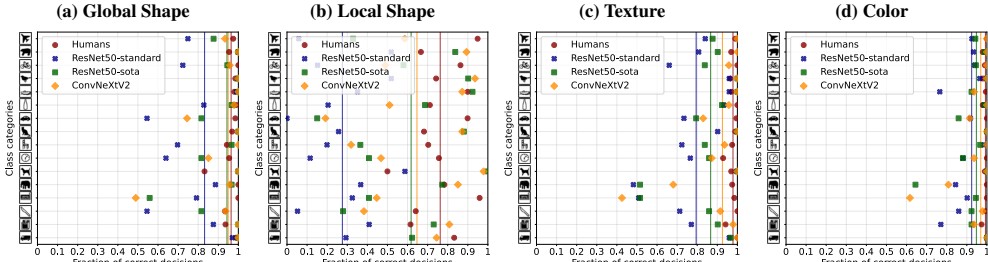

Figure 3: Relative accuracy under feature suppression for human observers and three CNNs ResNet50-standard, ResNet50-sota, ConvNeXtV2 on the curated ImageNet16 dataset. Each subplot shows performance under suppression of a specific feature: **(a)** global shape via Patch Shuffle (grid=3); **(b)** local shape via Patch Shuffle (grid=6); **(c)** texture via bilateral filtering; and **(d)** color via grayscale.

Table 3: Relative accuracy (accuracy under suppression divided by accuracy on original images) for each feature suppression type across models and human observers.

| Architecture | Global Shape | Local Shape | Texture | Color | Original | #Params |
|---|---|---|---|---|---|---|
| Humans | 0.965 | 0.763 | 0.979 | 0.999 | 0.969 | – |
| ResNet50-standard [2] | 0.832 | 0.276 | 0.795 | 0.924 | 0.954 | 25.6M |
| ResNet50-sota [46] | 0.943 | 0.618 | 0.867 | 0.948 | 0.931 | 25.6M |
| ConvNeXt [4] | 0.938 | 0.606 | 0.910 | 0.961 | 0.934 | 28.6M |
| ConvNeXtV2 [50] | 0.949 | 0.647 | 0.925 | 0.969 | 0.940 | 28.6M |
| EfficientNet [3] | 0.870 | 0.240 | 0.892 | 0.987 | 0.856 | 30.0M |
| EfficientNetV2 [48] | 0.926 | 0.423 | 0.897 | 0.957 | 0.932 | 24.0M |
| MobileNetV3 [47] | 0.795 | 0.217 | 0.761 | 0.859 | 0.881 | 5.4M |
| ConvMixer [49] | 0.920 | 0.437 | 0.815 | 0.891 | 0.874 | 21.1M |
| ViT [51] | 0.930 | 0.636 | 0.921 | 0.977 | 0.929 | 86.6M |
| DeiT [52] | 0.938 | 0.730 | 0.926 | 0.969 | 0.932 | 86.6M |
| Swin [53] | 0.924 | 0.713 | 0.906 | 0.941 | 0.945 | 87.8M |
| CLIP ViT [54] | 0.959 | 0.758 | 0.949 | 0.984 | 0.936 | 86.6M |

based models, the ViT demonstrates a feature reliance profile similar to ResNet50-sota across all suppression conditions, challenging the notion that transformers are inherently more shape-oriented than CNNs. Notably, the CLIP VIT model most closely matches human performance across all feature suppression conditions, suggesting that vision-language supervision encourages more human-aligned representations. This may reflect the effect of contrastive vision-language training, which prioritizes alignment with high-level semantic concepts over low-level visual cues.

These findings challenge the texture bias hypothesis popularized by Geirhos et al. [8] as a fixed inductive bias of CNNs. Instead, the observed behavior in the cue-conflict experiment may have reflected a dominant reliance on local shape features, rather than an inherent texture bias.

## 5.2 Experiment II: Domain-specific Feature Reliance

While Section 5.1 focuses on comparing feature reliance between humans and CNNs on a fixed benchmark, this section explores how reliance on shape, texture, and color varies across domains. The same suppression-based framework introduced earlier is applied to three representative visual domains: CV, MI, and RS. In each case, we fix the architecture to a ResNet50 and apply the standard training protocol, including only the data augmentation techniques random resized crop and horizontal flip. For CV datasets, we either train from scratch or initialize models with ImageNet-pretrained weights (standard training protocol) and then fine-tune on the respective datasets. For MI and RS, we train from scratch to allow a disentangled comparison across domains. Additional results for MI and RS with pretrained models to simulate operational scenarios can be found in the supplemental material. Details about the hyperparameter, as well as an overview of the corresponding validation accuracies, are provided in the supplemental material. In contrast to the previous experiment, in this

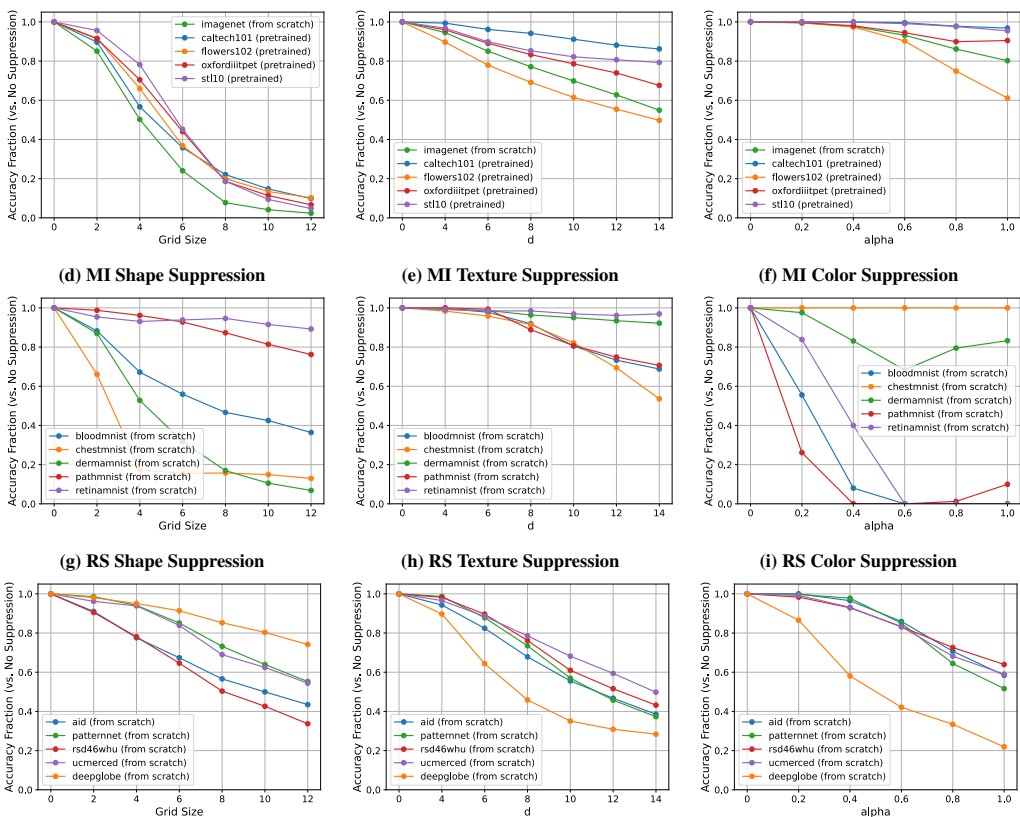

Figure 4: Feature suppression results across three domains. **Top row (a–c)**: ResNet50 pretrained on ImageNet and fine-tuned on CV datasets. **Middle row (d–f)**: ResNet50 trained from scratch on MI datasets from MedMNIST-v2. **Bottom row (g–i)**: ResNet50 trained from scratch on high-resolution RS datasets. Columns correspond to: **(a, d, g)** shape suppression (Patch Shuffle), **(b, e, h)** texture suppression (Bilateral Filter), and **(c, f, i)** color suppression (Grayscale).

experiment, suppression strength is treated as a continuous hyperparameter and systematically varied to obtain suppression curves that characterize feature reliance across domains. To reduce redundancy, we report results using one representative suppression technique per feature type in the main paper. Results using alternative suppression methods per feature type are included in the supplemental material and exhibit qualitatively similar patterns across domains.

To visualize domain-specific suppression sensitivity, we present a composite figure of per-domain results in Figure 4, showing the effect of suppressing shape, texture, and color for datasets from each domain. To ensure comparability across datasets with different numbers of classes and baseline accuracies, we standardize performance by rescaling: chance-level accuracy is mapped to 0, and baseline accuracy (i.e., accuracy on original images) is mapped to 1. Relative accuracy under suppression is then expressed on this normalized scale, facilitating direct comparison of feature reliance across domains and datasets. Finally, to synthesize the findings, we aggregate suppression curves in a domain-level comparison (Figure 5) by averaging results across datasets within each domain.

**Computer Vision (CV).** Figure 4a–c shows suppression results for five standard CV benchmarks (ImageNet [45], Caltech101 [56], Flowers102 [57], Oxford-IIIT-Pet [58], STL10 [59]). Across datasets, we observe that shape suppression induces the strongest performance degradation, especially as the patch shuffle grid size increases. This confirms a pronounced reliance on local shape information in pretrained CNNs. In contrast, texture suppression via bilateral filtering has minimal effect, and color suppression through grayscale conversion yields only minor degradation, indicating that CNNs fine-tuned on these datasets are largely robust to the removal of texture and color cues. These results

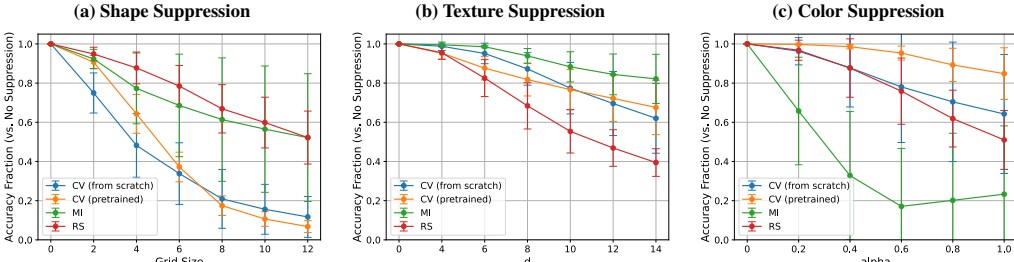

Figure 5: Domain-averaged feature suppression curves for CV, MI, and RS. **(a)** Shape suppression via Patch Shuffle. **(b)** Texture suppression via bilateral filtering. **(c)** Color suppression via grayscale.

are consistent with our human comparison study and suggest that local shape continues to dominate feature reliance in natural image classification tasks. For completeness, the supplemental material includes results for models trained from scratch as well as a class-wise analysis for ImageNet, which confirms that the global reliance patterns are consistent across categories.

**Medical Imaging (MI).** Figure 4d–f summarizes results on five datasets from the MedMNIST-v2 collection [60]: PathMNIST, RetinaMNIST, BloodMNIST, DermaMNIST, and ChestMNIST. We use the standardized $224 \times 224$ pixels version to ensure consistency with the experimental setup. Across these datasets, suppression effects are more heterogeneous than in CV. While shape suppression degrades performance, the impact is generally less pronounced, and texture suppression yields moderate performance drops in datasets such as PathMNIST and BloodMNIST, but relatively little effect in RetinaMNIST and DermaMNIST. By contrast, color suppression induces a substantial decline in classification accuracy for most datasets, reflecting the strong diagnostic role of chromatic cues, except in ChestMNIST, which contains only grayscale images. Taken together, these results suggest that feature reliance in MI varies substantially across datasets, with a common trend towards greater dependence on color information.

**Remote Sensing (RS).** Figure 4g–i reports suppression curves for five very-high-resolution RGB datasets: UCMerced [61], RSD46-WHU [62], DeepGlobe [63], PatternNet [64], and AID [65]. As in MI, shape suppression impacts performance, but the degradation is less pronounced than in the CV domain, indicating lower reliance on local shape. In contrast to CV and MI, texture suppression leads to substantial performance degradation across all datasets, suggesting that fine-grained surface patterns are critical for RS classification. Surprisingly, color suppression also results in notable performance drops, despite the use of RGB imagery only. This likely reflects strong correlations between chromatic cues and semantic land cover categories. Overall, RS models exhibit a pronounced reliance on texture and color, and comparatively less dependence on local shape, reflecting the distinct statistical structure and spatial semantics of RS imagery.

**Cross-Domain Comparison.** To synthesize these observations, Figure 5 presents the domain-averaged suppression curves for each feature, including 1-sigma error bars. Three clear trends emerge. First, CV models are most reliant on local shape, especially when trained from scratch, while ImageNet pretraining induces slightly greater robustness. Second, MI models exhibit stronger dependence on color, consistent with the nature of some medical tasks (e.g., in dermatology, histopathology), which often require interpreting chromatic cues. Third, RS models exhibit the highest texture reliance among the three tested domains. This may reflect the nature of many RS classes that are defined by texture-like patterns (e.g., fields, residential areas), rather than by distinct global contours. These patterns confirm that feature reliance is shaped not only by architecture and training regime, but also by the visual and semantic properties of the task or domain.

Finally, to validate the observed feature reliance patterns, we conduct complementary experiments on CV datasets with simultaneous suppression of two features (see Section I.5 in the supplemental material). Results confirm the trends of single-feature suppression: performance is highest when only shape is preserved, reduced when only texture remains, and nearly lost when only color is available. In summary, the findings highlight that domain characteristics, alongside architecture and training regime, play a crucial role in shaping feature reliance. While prior work emphasized architecture-induced biases, our results suggest that data properties equally govern the perceptual strategies that models adopt.

# 6 Conclusion

This paper revisited the widely cited claim that CNNs trained on ImageNet are inherently biased towards texture. We identify critical conceptual and methodological limitations in the cue-conflict experiment popularized by Geirhos et al. [8] that support this hypothesis. Further, we propose a new framework for evaluating feature reliance based on targeted suppression rather than forced-choice preference. Using this framework, we find no evidence for an inherent texture bias in CNNs, but instead observed a pronounced reliance on local shape features. Nonetheless, we show that this reliance can be substantially mitigated through modern training strategies. Across domains, we find that feature reliance varies substantially: CV models prioritize shape, MI models rely more evenly on color, and RS models exhibit strong texture sensitivity. These findings challenge the notion of fixed architectural biases and instead position feature reliance as a flexible property shaped by optimization objectives and domain-specific semantics, offering new directions for designing models that better align with human perceptual strategies. At the same time, the relative contributions of architectural components and training strategies to these reliance patterns remain to be systematically evaluated.

**Limitations.** Our framework relies on operational definitions of shape, texture, and color based on specific transformations, but features are continuous and interdependent, limiting perfect isolation. In practice, suppression only reduces rather than eliminates features: texture suppression can leave residual low-level features perceptible as texture, while shape suppression does not fully remove all shape cues. This reflects the inherent trade-off of reducing one feature while preserving others, making absolute removal unattainable. The applied suppression techniques may also introduce artifacts that affect model behavior independently of the targeted features (e.g., block-like structures from Patch Shuffle, smoothing from filtering). Further, the results obtained with pretrained models may reflect effects of similarities between suppression transformations and augmentation techniques (e.g., Cutout and Patch Shuffle). Finally, our human experiments employed a controlled forced-choice design with brief exposures and a limited set of categories to ensure comparability. While necessary for experimental control, these constraints may not fully reflect the richness and adaptability of human visual perception in real-world settings.

# Acknowledgements

We thank Johanna Vielhaben and Genc Hoxha for suggesting the addition of quantitative experiments to validate the feature suppression techniques, Liliann Lehrke for valuable guidance on the design of the human study, and Christopher Olk for helpful discussions in selecting the title. This work was partly supported by EU Horizon projects ELIAS (No. 101120237) and ELLIOT (No. 101214398).

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

# NeurIPS Paper Checklist

1. **Claims**

   Question: Do the main claims made in the abstract and introduction accurately reflect the paper's contributions and scope?

   Answer: [Yes] .

   Justification: The introduction and abstract clearly state the paper's core contributions: a critical re-evaluation of the cue-conflict protocol, the introduction of a suppression-based framework for assessing feature reliance, the empirical comparison of human and model reliance, and a further analysis across different visual domains. These claims are directly supported by the theoretical arguments in Section 3, the methodological framework in Section 4, and the empirical findings in Section 5.

   Guidelines:

   - The answer NA means that the abstract and introduction do not include the claims made in the paper.
   - The abstract and/or introduction should clearly state the claims made, including the contributions made in the paper and important assumptions and limitations. A No or NA answer to this question will not be perceived well by the reviewers.
   - The claims made should match theoretical and experimental results, and reflect how much the results can be expected to generalize to other settings.
   - It is fine to include aspirational goals as motivation as long as it is clear that these goals are not attained by the paper.

2. **Limitations**

   Question: Does the paper discuss the limitations of the work performed by the authors?

   Answer: [Yes] .

   Justification: The paper includes a dedicated paragraph for limitations at the end of the conclusion (Section 6), which acknowledges several key constraints. Specifically, it discusses the imperfect isolation of visual features due to their interdependent nature, potential artifacts introduced by suppression transformations (e.g., block structures or smoothing effects), and the constrained generalizability of the human study due to a controlled forced-choice setup with a limited category set and brief stimulus exposure.

   Guidelines:

   - The answer NA means that the paper has no limitation while the answer No means that the paper has limitations, but those are not discussed in the paper.
   - The authors are encouraged to create a separate "Limitations" section in their paper.
   - The paper should point out any strong assumptions and how robust the results are to violations of these assumptions (e.g., independence assumptions, noiseless settings, model well-specification, asymptotic approximations only holding locally). The authors should reflect on how these assumptions might be violated in practice and what the implications would be.
   - The authors should reflect on the scope of the claims made, e.g., if the approach was only tested on a few datasets or with a few runs. In general, empirical results often depend on implicit assumptions, which should be articulated.
   - The authors should reflect on the factors that influence the performance of the approach. For example, a facial recognition algorithm may perform poorly when image resolution is low or images are taken in low lighting. Or a speech-to-text system might not be used reliably to provide closed captions for online lectures because it fails to handle technical jargon.
   - The authors should discuss the computational efficiency of the proposed algorithms and how they scale with dataset size.
   - If applicable, the authors should discuss possible limitations of their approach to address problems of privacy and fairness.

- While the authors might fear that complete honesty about limitations might be used by reviewers as grounds for rejection, a worse outcome might be that reviewers discover limitations that aren't acknowledged in the paper. The authors should use their best judgment and recognize that individual actions in favor of transparency play an important role in developing norms that preserve the integrity of the community. Reviewers will be specifically instructed to not penalize honesty concerning limitations.

3. **Theory assumptions and proofs**

Question: For each theoretical result, does the paper provide the full set of assumptions and a complete (and correct) proof?

Answer: [NA] .

Justification: The paper does not include theoretical results.

Guidelines:

- The answer NA means that the paper does not include theoretical results.
- All the theorems, formulas, and proofs in the paper should be numbered and cross-referenced.
- All assumptions should be clearly stated or referenced in the statement of any theorems.
- The proofs can either appear in the main paper or the supplemental material, but if they appear in the supplemental material, the authors are encouraged to provide a short proof sketch to provide intuition.
- Inversely, any informal proof provided in the core of the paper should be complemented by formal proofs provided in appendix or supplemental material.
- Theorems and Lemmas that the proof relies upon should be properly referenced.

4. **Experimental result reproducibility**

Question: Does the paper fully disclose all the information needed to reproduce the main experimental results of the paper to the extent that it affects the main claims and/or conclusions of the paper (regardless of whether the code and data are provided or not)?

Answer: [Yes] .

Justification: The paper provides sufficient methodological detail to reproduce the main experimental results. It specifies all datasets used (e.g., ImageNet16, MedMNIST, RS benchmarks), feature suppression transformations with hyperparameters (e.g., Patch Shuffle grid sizes, bilateral filter settings), model architectures (including training recipes and pretrained sources), and evaluation procedures for both human and model experiments. For the human study, participant design, timing protocols, category sets, and interface descriptions are clearly outlined. Additional implementation details, ablations, and visual examples are provided in the supplemental material. Together, these descriptions enable reproduction of the core results and validation of the paper's main claims. Upon acceptance, our code will be made publicly available,e containing the necessary scripts or CLI commands to reproduce the experiments.

Guidelines:

- The answer NA means that the paper does not include experiments.
- If the paper includes experiments, a No answer to this question will not be perceived well by the reviewers: Making the paper reproducible is important, regardless of whether the code and data are provided or not.
- If the contribution is a dataset and/or model, the authors should describe the steps taken to make their results reproducible or verifiable.
- Depending on the contribution, reproducibility can be accomplished in various ways. For example, if the contribution is a novel architecture, describing the architecture fully might suffice, or if the contribution is a specific model and empirical evaluation, it may be necessary to either make it possible for others to replicate the model with the same dataset, or provide access to the model. In general. releasing code and data is often one good way to accomplish this, but reproducibility can also be provided via detailed instructions for how to replicate the results, access to a hosted model (e.g., in the case of a large language model), releasing of a model checkpoint, or other means that are appropriate to the research performed.

- While NeurIPS does not require releasing code, the conference does require all submissions to provide some reasonable avenue for reproducibility, which may depend on the nature of the contribution. For example

    (a) If the contribution is primarily a new algorithm, the paper should make it clear how to reproduce that algorithm.

    (b) If the contribution is primarily a new model architecture, the paper should describe the architecture clearly and fully.

    (c) If the contribution is a new model (e.g., a large language model), then there should either be a way to access this model for reproducing the results or a way to reproduce the model (e.g., with an open-source dataset or instructions for how to construct the dataset).

    (d) We recognize that reproducibility may be tricky in some cases, in which case authors are welcome to describe the particular way they provide for reproducibility. In the case of closed-source models, it may be that access to the model is limited in some way (e.g., to registered users), but it should be possible for other researchers to have some path to reproducing or verifying the results.

5. **Open access to data and code**

   Question: Does the paper provide open access to the data and code, with sufficient instructions to faithfully reproduce the main experimental results, as described in supplemental material?

   Answer: [Yes] .

   Justification: Upon acceptance, our code will be made publicly available, which contains the full implementation of the proposed suppression-based evaluation framework, scripts for reproducing all key experiments, and instructions for environment setup and execution. The code also contains the tool for the human study interface. All datasets used are publicly available, and instructions for accessing them are provided.

   Guidelines:

   - The answer NA means that paper does not include experiments requiring code.
   - Please see the NeurIPS code and data submission guidelines (`https://nips.cc/public/guides/CodeSubmissionPolicy`) for more details.
   - While we encourage the release of code and data, we understand that this might not be possible, so "No" is an acceptable answer. Papers cannot be rejected simply for not including code, unless this is central to the contribution (e.g., for a new open-source benchmark).
   - The instructions should contain the exact command and environment needed to run to reproduce the results. See the NeurIPS code and data submission guidelines (`https://nips.cc/public/guides/CodeSubmissionPolicy`) for more details.
   - The authors should provide instructions on data access and preparation, including how to access the raw data, preprocessed data, intermediate data, and generated data, etc.
   - The authors should provide scripts to reproduce all experimental results for the new proposed method and baselines. If only a subset of experiments are reproducible, they should state which ones are omitted from the script and why.
   - At submission time, to preserve anonymity, the authors should release anonymized versions (if applicable).
   - Providing as much information as possible in supplemental material (appended to the paper) is recommended, but including URLs to data and code is permitted.

6. **Experimental setting/details**

   Question: Does the paper specify all the training and test details (e.g., data splits, hyperparameters, how they were chosen, type of optimizer, etc.) necessary to understand the results?

   Answer: [Yes] .

   Justification: The paper provides all essential training and test details necessary to understand the experimental results. It describes dataset preprocessing, suppression methods with fixed parameters, and evaluation protocols. For model experiments, it distinguishes

between models trained from scratch and those finetuned from pretrained checkpoints (e.g., from `timm`), and explicitly describes the timm training augmentations, optimizers, and architecture-specific configurations. Details such as how suppression conditions were balanced across human participants and models are given in the main text, while additional hyperparameter settings and ablations are included in the supplemental material.

Guidelines:

- The answer NA means that the paper does not include experiments.
- The experimental setting should be presented in the core of the paper to a level of detail that is necessary to appreciate the results and make sense of them.
- The full details can be provided either with the code, in appendix, or as supplemental material.

7. **Experiment statistical significance**

Question: Does the paper report error bars suitably and correctly defined or other appropriate information about the statistical significance of the experiments?

Answer: [Yes] .

Justification: For the human vs. CNN comparison (Figure 3, Table 3), statistical significance is assessed using paired t-tests between human and model performance under each suppression condition and is presented in the supplemental material. For the domain comparison experiments (Figure 5), 1-sigma error bars are reported across datasets within each domain to reflect inter-dataset variability in suppression sensitivity.

Guidelines:

- The answer NA means that the paper does not include experiments.
- The authors should answer "Yes" if the results are accompanied by error bars, confidence intervals, or statistical significance tests, at least for the experiments that support the main claims of the paper.
- The factors of variability that the error bars are capturing should be clearly stated (for example, train/test split, initialization, random drawing of some parameter, or overall run with given experimental conditions).
- The method for calculating the error bars should be explained (closed form formula, call to a library function, bootstrap, etc.)
- The assumptions made should be given (e.g., Normally distributed errors).
- It should be clear whether the error bar is the standard deviation or the standard error of the mean.
- It is OK to report 1-sigma error bars, but one should state it. The authors should preferably report a 2-sigma error bar than state that they have a 96% CI, if the hypothesis of Normality of errors is not verified.
- For asymmetric distributions, the authors should be careful not to show in tables or figures symmetric error bars that would yield results that are out of range (e.g. negative error rates).
- If error bars are reported in tables or plots, The authors should explain in the text how they were calculated and reference the corresponding figures or tables in the text.

8. **Experiments compute resources**

Question: For each experiment, does the paper provide sufficient information on the computer resources (type of compute workers, memory, time of execution) needed to reproduce the experiments?

Answer: [Yes] .

Justification: We dedicated a section in the supplemental material to hardware specifications, memory and runtime.

Guidelines:

- The answer NA means that the paper does not include experiments.
- The paper should indicate the type of compute workers CPU or GPU, internal cluster, or cloud provider, including relevant memory and storage.

- The paper should provide the amount of compute required for each of the individual experimental runs as well as estimate the total compute.
- The paper should disclose whether the full research project required more compute than the experiments reported in the paper (e.g., preliminary or failed experiments that didn't make it into the paper).

9. **Code of ethics**

Question: Does the research conducted in the paper conform, in every respect, with the NeurIPS Code of Ethics https://neurips.cc/public/EthicsGuidelines?

Answer: [Yes] .

Justification: The research adheres to the NeurIPS Code of Ethics. All datasets used are publicly available. The human study was conducted in accordance with institutional ethical guidelines. All participants provided informed consent prior to participation. The experiments do not involve sensitive data, privacy risks, or unfair bias against individuals or groups. Results are reported transparently, with limitations and assumptions clearly stated.

Guidelines:

- The answer NA means that the authors have not reviewed the NeurIPS Code of Ethics.
- If the authors answer No, they should explain the special circumstances that require a deviation from the Code of Ethics.
- The authors should make sure to preserve anonymity (e.g., if there is a special consideration due to laws or regulations in their jurisdiction).

10. **Broader impacts**

Question: Does the paper discuss both potential positive societal impacts and negative societal impacts of the work performed?

Answer: [NA] .

Justification: The paper presents foundational research on evaluating feature reliance in image classifiers through controlled suppression. It does not introduce application-specific systems or deployment scenarios that would entail direct societal impact.

Guidelines:

- The answer NA means that there is no societal impact of the work performed.
- If the authors answer NA or No, they should explain why their work has no societal impact or why the paper does not address societal impact.
- Examples of negative societal impacts include potential malicious or unintended uses (e.g., disinformation, generating fake profiles, surveillance), fairness considerations (e.g., deployment of technologies that could make decisions that unfairly impact specific groups), privacy considerations, and security considerations.
- The conference expects that many papers will be foundational research and not tied to particular applications, let alone deployments. However, if there is a direct path to any negative applications, the authors should point it out. For example, it is legitimate to point out that an improvement in the quality of generative models could be used to generate deepfakes for disinformation. On the other hand, it is not needed to point out that a generic algorithm for optimizing neural networks could enable people to train models that generate Deepfakes faster.
- The authors should consider possible harms that could arise when the technology is being used as intended and functioning correctly, harms that could arise when the technology is being used as intended but gives incorrect results, and harms following from (intentional or unintentional) misuse of the technology.
- If there are negative societal impacts, the authors could also discuss possible mitigation strategies (e.g., gated release of models, providing defenses in addition to attacks, mechanisms for monitoring misuse, mechanisms to monitor how a system learns from feedback over time, improving the efficiency and accessibility of ML).

11. **Safeguards**

Question: Does the paper describe safeguards that have been put in place for responsible release of data or models that have a high risk for misuse (e.g., pretrained language models, image generators, or scraped datasets)?

Answer: [NA] .

Justification: The paper does not release any new models or datasets with a high risk of misuse. All experiments are conducted using publicly available models (e.g., from `timm`) and datasets that are standard in the community and pose no known safety concerns. The feature suppression framework is evaluation-focused and not inherently dual-use.

Guidelines:

- The answer NA means that the paper poses no such risks.
- Released models that have a high risk for misuse or dual-use should be released with necessary safeguards to allow for controlled use of the model, for example by requiring that users adhere to usage guidelines or restrictions to access the model or implementing safety filters.
- Datasets that have been scraped from the Internet could pose safety risks. The authors should describe how they avoided releasing unsafe images.
- We recognize that providing effective safeguards is challenging, and many papers do not require this, but we encourage authors to take this into account and make a best faith effort.

12. **Licenses for existing assets**

Question: Are the creators or original owners of assets (e.g., code, data, models), used in the paper, properly credited and are the license and terms of use explicitly mentioned and properly respected?

Answer: [Yes] .

Justification: All datasets used in this work are publicly available and distributed under open licenses. All software packages, models, and datasets used are properly cited in the paper with references to their original sources and licenses where applicable.

Guidelines:

- The answer NA means that the paper does not use existing assets.
- The authors should cite the original paper that produced the code package or dataset.
- The authors should state which version of the asset is used and, if possible, include a URL.
- The name of the license (e.g., CC-BY 4.0) should be included for each asset.
- For scraped data from a particular source (e.g., website), the copyright and terms of service of that source should be provided.
- If assets are released, the license, copyright information, and terms of use in the package should be provided. For popular datasets, paperswithcode.com/datasets has curated licenses for some datasets. Their licensing guide can help determine the license of a dataset.
- For existing datasets that are re-packaged, both the original license and the license of the derived asset (if it has changed) should be provided.
- If this information is not available online, the authors are encouraged to reach out to the asset's creators.

13. **New assets**

Question: Are new assets introduced in the paper well documented and is the documentation provided alongside the assets?

Answer: [Yes] .

Justification: The paper introduces a new evaluation framework based on targeted feature suppression and provides accompanying code and configuration files for reproducibility. The supplemental material includes the usage instructions, dataset preprocessing steps, transformation parameters, and experiment scripts. No new datasets or models are released, and all components are based on existing, publicly available assets.

Guidelines:

- The answer NA means that the paper does not release new assets.

- Researchers should communicate the details of the dataset/code/model as part of their submissions via structured templates. This includes details about training, license, limitations, etc.
- The paper should discuss whether and how consent was obtained from people whose asset is used.
- At submission time, remember to anonymize your assets (if applicable). You can either create an anonymized URL or include an anonymized zip file.

14. **Crowdsourcing and research with human subjects**

Question: For crowdsourcing experiments and research with human subjects, does the paper include the full text of instructions given to participants and screenshots, if applicable, as well as details about compensation (if any)?

Answer: [Yes] .

Justification: The paper includes a detailed description of the experimental procedure for the human study in the main text and provides the full participant instructions and interface screenshots in the supplemental material. Participants were volunteers and were not financially compensated. The study followed institutional ethical guidelines, and informed consent was obtained from all participants.

Guidelines:

- The answer NA means that the paper does not involve crowdsourcing nor research with human subjects.
- Including this information in the supplemental material is fine, but if the main contribution of the paper involves human subjects, then as much detail as possible should be included in the main paper.
- According to the NeurIPS Code of Ethics, workers involved in data collection, curation, or other labor should be paid at least the minimum wage in the country of the data collector.

15. **Institutional review board (IRB) approvals or equivalent for research with human subjects**

Question: Does the paper describe potential risks incurred by study participants, whether such risks were disclosed to the subjects, and whether Institutional Review Board (IRB) approvals (or an equivalent approval/review based on the requirements of your country or institution) were obtained?

Answer: [Yes] .

Justification: A detailed description of the risk assessment and consent procedures is provided in the supplemental material. The study involved a low-risk visual classification task with adult participants, with no foreseeable harm. All participants were volunteers and provided informed consent. The study followed the ethical protocols of our institution and adhered to all relevant institutional guidelines. A review was conducted in accordance with our institution's standard ethics procedures, and no ethical concerns were identified.

Guidelines:

- The answer NA means that the paper does not involve crowdsourcing nor research with human subjects.
- Depending on the country in which research is conducted, IRB approval (or equivalent) may be required for any human subjects research. If you obtained IRB approval, you should clearly state this in the paper.
- We recognize that the procedures for this may vary significantly between institutions and locations, and we expect authors to adhere to the NeurIPS Code of Ethics and the guidelines for their institution.
- For initial submissions, do not include any information that would break anonymity (if applicable), such as the institution conducting the review.

16. **Declaration of LLM usage**

Question: Does the paper describe the usage of LLMs if it is an important, original, or non-standard component of the core methods in this research? Note that if the LLM is used

only for writing, editing, or formatting purposes and does not impact the core methodology, scientific rigorousness, or originality of the research, declaration is not required.

Answer: [NA] .

Justification: The core method development in this research does not involve LLMs as any important, original, or non-standard components.

Guidelines:

- The answer NA means that the core method development in this research does not involve LLMs as any important, original, or non-standard components.
- Please refer to our LLM policy (`https://neurips.cc/Conferences/2025/LLM`) for what should or should not be described.

