# OpenReview forum: "ImageNet-trained CNNs are not biased towards texture: Revisiting feature reliance through controlled suppression"
_NeurIPS.cc/2025/Conference — NeurIPS 2025 oral_

### Official Review · Reviewer_rbD7 · 2025-06-27

**Clarity:** 4
**Significance:** 3
**Originality:** 3
**Rating:** 5
**Confidence:** 3

**Summary:**

Previous work empirically establishes that CNNs are biased towards textures. This paper offers a critique on the nature and setting of the experiments presented in previous work. It also offers a new experimental methodology to show that CNNs and Transformers are not biased towards textures, but like humans, they are biased towards local shapes, at least in the computer vision domain.

Three feature types are identified: shape, texture, and color. Several feature suppression transformations for each type are proposed and validated. Experiments evaluate the classification accuracy, under feature suppression, for human behaviour and several models. Additional experiments evaluate and compare different domains.

Models with similar architectures that employed more involved training strategies, such as data augmentation, showed less bias towards local shape. Results show that the medical imaging domain is biased towards color, while remote sensing models are more affected by changes in texture.

**Questions:**

Could you please provide reasoning that might help me reduce any of the four concerns I have raised as weaknesses?

**Ethical Concerns:**

["NO or VERY MINOR ethics concerns only"]

**Final Justification:**

I no longer have concerns about the interdependence of feature types. I believe the paper will be much stronger with the additional results proposed by the authors. As for the other concerns, I think addressing them as suggested will be a great idea.

Although I agree with the comments in the rebuttal and I am not as concerned about the points raised, I am still not completely satisfied. It seems to me like we are retaining more texture information than other types of information. This is a completely subjective opinion of course, and that is what leads me to the next concern: what is preventing this work from being criticized in the future and replaced by a different methodology that arrives at a different conclusion? As a counterargument: this methodology is definitely much more solid than previous work.

I will keep my overall score but I will raise the quality to excellent and originality to good.

**Limitations:**

Yes, limitations are adequately addressed. Perhaps some additional considerations related to the concerns raised in this review should have been mentioned.

**Paper Formatting Concerns:**

Not an issue but Figure 3 would be better placed on the previous page.

**Quality:**

4

**Strengths And Weaknesses:**

Strengths:
1. Results provide interesting and actionable insights
2. Quantitative validation of suppression transformations helps justify the experimental methodology
3. Statistical significance tests increases the strength of the claim
4. Thorough coverage of different model types and domains

Weaknesses:
1. Observing the visual examples in the Appendix, the proposed suppression transformations for texture still retain low frequency information, which, to the human eye at least, seem to still give away a lot of texture. The LV and HFE metrics, understandably, do not seem to capture this
2. The causes for the suggested inductive biases are not explained. As shown in some cases, if the architecture (and not the nature of the task domain or the lack of an adequate training strategy) causes some of the bias, which characteristics cause it? This analysis might seem outside of the scope of this work, but in its absence, the validity of the methodology takes more importance and I am concerned that, similarly to the previous work, it might not have been validated enough
3. To which extent do the data augmentations resemble the suppression transformations used? If we are training models using similar transformations, then we might just be overfitting to them. Thus, other suppression techniques might yield very different results. In particular, the Mixup, Cutout, and Cutmix augmentations can resemble some of the transformations
4. As pointed out in the limitations, feature types seem to be interdependent. What if the observed biases only exist in the presence of other features. For example, what if local shapes are not as relevant in the absence of texture or color? Such behavior might produce different conclusions

---

> ### Author Rebuttal · Authors · 2025-07-29
>
> We thank the reviewer for their thoughtful and constructive feedback, as well as for their careful reading of our work. We particularly appreciate the recognition of our methodological contributions and the detailed suggestions regarding limitations and interpretation. We respond to each of the points raised below.
>
> **Visual examples of texture suppression still exhibit some degree of texture.**
> We acknowledge the reviewer’s observation that the proposed texture suppression transformations, while quantitatively validated, may retain some degree of low-level visual features that could still be perceived as texture by the human eye. This is consistent with the fact that our suppression metrics (e.g., LV and HFE) do not approach zero, reflecting a deliberate trade-off: suppression strength was selected to reduce texture while preserving shape, particularly to avoid compromising classification signals through over-smoothing. Notably, this limitation is symmetric: shape suppression (e.g., via PatchShuffle) also does not fully eliminate all shape information. Absolute suppression of individual feature types is rarely achievable due to their interdependence, and our goal is not to remove a feature entirely but to degrade it sufficiently to observe relative reliance. We will make this point more explicit in the limitations section in L325.
>
> **The causes for the suggested inductive biases are not explained.**
> While this work focuses on quantifying feature reliance rather than identifying its underlying causes, we acknowledge the importance of understanding what drives the observed differences across models. As the reviewer notes, architectural characteristics (e.g., convolutional locality, depth, attention mechanisms) and training strategies (e.g., augmentations, regularization) likely contribute in complex ways to feature reliance. A more systematic ablation study, isolating specific architectural components and optimization choices, would be needed to identify these causal factors more precisely. We consider this an important direction for future work that we will add to our conclusion in L320. That said, our current results already suggest that reliance tendencies are not rigid architectural properties: ConvNeXtV2, a convolutional model, exhibits reliance patterns more aligned with transformers and vision-language models, highlighting the significant role of training procedures.
>
> **To which extent do the data augmentations resemble the suppression transformations used? If we are training models using similar transformations, then we might just be overfitting to them.**
> We thank the reviewer for raising an important limitation that was not previously discussed. Some suppression transformations, particularly those targeting shape (e.g., PatchShuffle), may share structural similarities with augmentation techniques such as Cutout, CutMix or other geometric data augmentation techniques. This resemblance could lead to inflated robustness in models trained with such augmentations, not due to generalized feature use but because of exposure to similar distortions during training. To mitigate this risk, we deliberately excluded such augmentation strategies from all training pipelines for models trained from scratch. Advanced augmentation techniques are only present in models imported from external sources (e.g., pretrained variants from the timm library), where the training protocol was not under our direct control. While we believe that the exact degree of overlap is difficult to quantify, we acknowledge this as a valid concern and will include it as an explicit limitation in L325.
>
> **Interdependence of feature types: What if the observed biases only exist in the presence of other features?**
> The potential interdependence between feature types is a valuable consideration and suggests a meaningful extension of our evaluation. To assess whether the observed feature reliance patterns persist when multiple features are suppressed simultaneously, we conducted an additional experiment using joint suppression for the CV datasets.
>
> Specifically, we tested three suppression combinations: color + texture (leaving only shape), color + shape (leaving only texture), and shape + texture (leaving only color). For each case, we computed the accuracy fraction as a function of increasing suppression strength, analogous to the plots shown in Figure 4. Due to NeurIPS 2025 constraints, we summarize the results only for ImageNet in the table below (in the Figure all datasets are evaluated), reporting the model's accuracy fraction at each suppression level:
>
> | Suppression Type / strength         | 0     | 2     | 4     | 6     | 8     | 10    | 12    |
> |-------------------------------------|-------|-------|-------|-------|-------|-------|-------|
> | Color + Texture (only Shape preserved)   | 1.000 | 0.714 | 0.592 | 0.511 | 0.451 | 0.396 | 0.339 |
> | Color + Shape   (only Texture preserved) | 1.000 | 0.578 | 0.219 | 0.079 | 0.026 | 0.015 | 0.010 |
> | Shape + Texture (only Color preserved)   | 1.000 | 0.415 | 0.121 | 0.043 | 0.016 | 0.010 | 0.007 |
>
> These results confirm and extend the patterns seen in single-feature suppression: accuracy is the highest when only shape is preserved, lower when only texture is available, and nearly lost when only color remains. This supports the interpretation that feature reliance remains consistent even when features are suppressed in combination. While these values are reported for ImageNet, the final version of the paper will include corresponding figures for all computer vision datasets in the supplemental material.

---

### Official Review · Reviewer_mrhF · 2025-06-30

**Clarity:** 4
**Significance:** 4
**Originality:** 4
**Rating:** 6
**Confidence:** 4

**Summary:**

The paper revisits the "texture bias" hypothesis introduced by Geirhos et al. CNNs preferentially rely on texture over shape. The authors argue that the classic cue‐conflict evaluation (combining one class's shape with another's texture via neural style transfer) conflates salience with reliance and entangles multiple features. To address these limitations, they propose a domain-agnostic suppression framework: for each feature class (shape, texture, colour), two targeted image transformations systematically suppress that feature while preserving others. They validate these transformations quantitatively.

Using this framework, they conduct two large‐scale experimental studies:

1. Human vs. CNN feature reliance on the ImageNet16 dataset (50 images × 16 classes), comparing human observers to a range of CNNs (ResNets, EfficientNets, ConvNeXts) and transformers (ViT, Swin, CLIP-ViT). They show that (a) all models—including humans—are most vulnerable to local shape suppression, not texture; (b) modern training recipes and vision-language supervision mitigate this reliance; and (c) CNNs are not inherently texture-biased.

2. Cross-domain analysis on computer-vision (ImageNet, Flowers, Pets, etc.), medical imaging (MedMNIST-v2), and remote-sensing (UCMerced, DeepGlobe, etc.). They find that CV models prioritise shape, MI models lean on colour, and RS models depend heavily on texture.

Overall, the paper challenges a prevailing narrative, presents a clear empirical protocol, and sheds light on how feature reliance varies across different architectures, training strategies, and domains.

**Questions:**

- **Thresholding rationale (line 206):** Why choose a 0.5 softmax threshold for correctness? How do results differ if you use argmax instead?

- **Visualisations for MI and RS samples:** You include two ImageNet examples of suppressed images; please provide analogous visualisations for all medical imaging and remote sensing datasets as well. They need not match the scale of the ImageNet figures but should illustrate key transformations in those domains.

- **Attribution to earlier work for the patch-shuffle idea for analysing shape bias** The patch‐shuffle technique used to disrupt shape traces back to earlier work [1], which further cites [2] for the idea. The current manuscript lacks explicit attribution to either papers or any other prior work that first introduced the use of the patch-shuffle idea for shape analysis.

- **Effect of patch rotation:** How much effect does patch rotation have on suppressing shape? How much do results vary if you shuffle the patches but do not rotate them? My guess would be that patch rotation has a larger effect when the grid size is small (3x3) but a smaller effect when it's large (6x6).

**[1]** Mummadi, Chaithanya Kumar, et al. "Does enhanced shape bias improve neural network robustness to common corruptions?." arXiv preprint arXiv:2104.09789 (2021).

**[2]** Luo, Tiange, et al. "Defective Convolutional Networks." arXiv preprint arXiv:1911.08432 (2019).

**Ethical Concerns:**

["NO or VERY MINOR ethics concerns only"]

**Final Justification:**

I commend the authors on their thorough rebuttal and the additional experiments, which satisfactorily address all of my initial concerns and fully clarify the questions I raised. Moreover, having reviewed the other reviewers’ comments, I find no outstanding issues or unresolved critiques. Consequently, I am confident that the manuscript now meets all expectations, and I hereby raise my score to 6.

**Limitations:**

yes

**Quality:**

3

**Strengths And Weaknesses:**

### **Strengths:**
- **Novelty of protocol:** The proposed protocol of targeted feature suppression yields a more direct and generalizable measure of reliance than the shape-cue conflict protocol.

- Human evaluation is used as a baseline, and comparisons are made with various CNN and transformer architectures.

- Analysis extends beyond natural images to medical imaging and remote sensing, revealing domain-specific reliance patterns.

- **Clarity of writing and organisation:** The paper is well–structured and easy to read. It has clear figures, tables, and supplemental material.

### **Weaknesses:**
- **Unrealistic training setup in specialised domains:** The authors train ResNet-50 from scratch on Medical Imaging and Remote Sensing datasets to “reflect realistic usage in specialised domains” (lines 250–251). However, in histopathology (e.g., PathMNIST), researchers almost invariably initialise CNNs with ImageNet-pretrained weights. In Multiple Instance Learning (MIL), practitioners typically use a frozen ResNet-50 (initialised with ImageNet weights) to extract embeddings. Moreover, realistic workflows employ not only geometric augmentations but also photometric variants (e.g., ColorJitter), which can reduce colour bias. While it's common to not use any data augmentation when using a frozen backbone in MIL (mostly to save GPU memory), other setups do apply it. From what I could find, it is generally common to fine-tune ImageNet-pretrained models with extensive data augmentation in both MI and RS domains; thus, experimenting under that paradigm would be better.

- **Lack of class-wise analysis in Experiment II:** For the experiments in Experiment II, especially ImageNet-1K, a more fine-grained, per-class evaluation (or reporting the proportion of classes supporting the hypothesis of reliance on local shape over texture) would clarify whether this effect holds uniformly across all categories or only emerges in a subset.

---

> ### Author Rebuttal · Authors · 2025-07-29
>
> We thank the reviewer for their detailed and insightful comments. Below, we provide a point-by-point response addressing all raised questions and suggestions.
>
> **Unrealistic training setup in specialised domains:**
> We thank the reviewer for this valuable clarification. We agree that our phrasing regarding “realistic usage in specialized domains” (L250–251) was too strong. Our primary motivation for training from scratch in medical imaging (MI) and remote sensing (RS) was to allow a disentangled comparison across domains, avoiding potential reliance profiles introduced by ImageNet pretraining. However, we fully acknowledge that in many operational workflows, practitioners in both domains often initialize from ImageNet-pretrained backbones, especially in histopathology (e.g., PathMNIST) or MIL settings. Hence, we will revise the mentioned section to “For MI and RS, we train from scratch to allow a disentangled comparison across domains. Additional results for MI and RS with pre-trained models to simulate operational scenarios can be found in the supplemental material”.
>
> To address this point, we followed the reviewer’s suggestion and repeated the domain-specific suppression experiments with ImageNet-pretrained ResNet-50 backbones for both domains. The setup and suppression protocols follow those of Figure 4 in the main paper. As a proxy for the new plots that we will integrate into the final version, the following three tables report accuracy degradation under increasing suppression strength for shape, texture, and color suppression, respectively. Each table compares performance for models trained from scratch (first value in each cell) versus pretrained ImageNet weights fine-tuned on the datasets (second value in each cell). The overall reliance patterns remain broadly consistent. However, as observed in the CV domain, ImageNet pretraining tends to subtly increase shape reliance and decrease reliance on texture and color in RS (e.g., up to 7% stronger shape reliance, up to 20% less texture/color reliance). In MI, pretraining yields mixed effects: for BloodMNIST it increases shape reliance under strong suppression (e.g., by 20% for grid-size 8), whereas for PathMNIST it amplifies sensitivity to texture suppression. This reinforces the importance of considering both training paradigms when analyzing feature reliance, and we will ensure that this is clearly reflected in the revised manuscript.
>
> **Shape Suppression**
>
> | Dataset      | 2           | 4           | 6           | 8           | 10          |
> |--------------|-------------|-------------|-------------|-------------|-------------|
> | PatternNet   | 0.99 / 0.99 | 0.94 / 0.93 | 0.85 / 0.83 | 0.73 / 0.69 | 0.64 / 0.58 |
> | AID          | 0.91 / 0.95 | 0.78 / 0.79 | 0.67 / 0.66 | 0.57 / 0.52 | 0.50 / 0.43 |
> |          |          |          |          |          |          |
> | BloodMNIST   | 0.88 / 0.93 | 0.67 / 0.83 | 0.56 / 0.65 | 0.47 / 0.26 | 0.42 / 0.04 |
> | PathMNIST    | 0.99 / 1.00 | 0.96 / 0.98 | 0.93 / 0.95 | 0.87 / 0.91 | 0.81 / 0.86 |
>
> **Texture Suppression**
>
> | Dataset      | 4           | 6           | 8           | 10          | 12          |
> |--------------|-------------|-------------|-------------|-------------|-------------|
> | PatternNet   | 0.99 / 0.99 | 0.88 / 0.92 | 0.74 / 0.80 | 0.57 / 0.69 | 0.46 / 0.58 |
> | AID          | 0.94 / 0.96 | 0.82 / 0.90 | 0.68 / 0.82 | 0.56 / 0.69 | 0.47 / 0.58 |
> |          |          |          |          |          |          |
> | BloodMNIST   | 1.00 / 0.95 | 0.98 / 0.83 | 0.92 / 0.73 | 0.81 / 0.61 | 0.73 / 0.52 |
> | PathMNIST    | 1.00 / 1.00 | 0.99 / 0.94 | 0.89 / 0.82 | 0.81 / 0.74 | 0.75 / 0.70 |
>
> **Color Suppression**
>
> | Dataset      | 0.2         | 0.4         | 0.6         | 0.8         | 1.0         |
> |--------------|-------------|-------------|-------------|-------------|-------------|
> | PatternNet   | 1.00 / 1.00 | 0.98 / 0.99 | 0.85 / 0.90 | 0.64 / 0.66 | 0.52 / 0.53 |
> | AID          | 1.00 / 1.00 | 0.97 / 0.99 | 0.86 / 0.94 | 0.71 / 0.82 | 0.58 / 0.73 |
> |          |          |          |          |          |          |            |
> | BloodMNIST   | 0.56 / 0.70 | 0.08 / 0.23 | 0.00 / 0.00 | 0.00 / 0.00 | 0.00 / 0.00 |
> | PathMNIST    | 0.26 / 0.94 | 0.00 / 0.82 | 0.00 / 0.49 | 0.01 / 0.30 | 0.10 / 0.21 |
>
> **Class-wise analysis for ImageNet in Experiment II:**
> We thank the reviewer for this valuable suggestion. We have now conducted this analysis to examine whether the observed feature reliance patterns in Experiment II (specifically on ImageNet-1K) generalize across classes or are driven by a subset.
>
> To this end, we evaluated class-wise performance under local shape suppression (patch-shuffle, grid size 6), texture suppression (bilateral filter, kernel size 12) and color suppression. We computed the relative accuracy for each of the 1,000 ImageNet classes under the suppression conditions and visualized the distributions using kernel density estimates (KDEs). The class-level distributions confirm the global trend: the KDE for texture-suppressed scores is clearly shifted toward higher values (mean = 0.62) than that for local-shape-suppressed scores (mean = 0.24), with only modest overlap between the two distributions (approximately 25–30%).
>
> To quantify class consistency, we computed the proportion of classes for which the relative feature importance followed the global pattern. We find:
> - 88.0% of classes exhibit greater local shape than texture reliance
> - 94.2% exhibit greater local shape than color reliance
> - 77.6% exhibit greater texture than color reliance
>
> This supports that the effect is not limited to a few outlier categories but holds broadly across the dataset.
>
> To complement the quantitative analysis, we also identified representative outlier classes. Classes where the model exhibits unusually low reliance on local shape include corn, Appenzeller (cheese), bookstall, spider's web, zebra, guacamole, and stone wall. Classes with unusually strong texture reliance include Chesapeake Bay retriever, Crotalus cerastes, stingray, bath towel, Alligator mississippiensis, and bolete. For color reliance, outliers include sunglass, tank suit, ladle, coffeepot, chain, and tam-tam.
>
> We will include the KDE plot and the results of this class-wise analysis in the final version to provide a more granular understanding of per-class feature reliance.
>
> **Thresholding rationale (line 206):**
> In our main experiments, we applied a 0.5 softmax threshold to determine correctness. This was chosen heuristically to ensure that predictions reflect not only the top class but also a minimal level of confidence, particularly under suppression. We acknowledge that this introduces an implicit confidence calibration assumption.
>
> To assess the sensitivity of our results to this threshold, we re-computed the scores using a plain argmax decision rule, which always selects the top-scoring class regardless of confidence. As expected, absolute accuracies are slightly higher under argmax, but the relative pattern of degradation across suppression types and models remains qualitatively consistent, supporting the robustness of our findings.
>
> Below we report both versions side by side for three representative models:
> | Model Variant                      | Global Shape | Local Shape | Texture | Color  | Original |
> | --- | --- | --- | --- | --- | --- |
> | **ResNet50-standard (Softmax > 0.5)** | 0.832        | 0.276       | 0.795   | 0.924  | 0.954    |
> | **ResNet50-standard (Argmax)**        | 0.880        | 0.361       | 0.840   | 0.944  | 0.962    |
> |--------------------------------|--------|-------|-------|-------|-------|
> | **ResNet50-sota (Softmax > 0.5)**     | 0.943        | 0.618       | 0.867   | 0.948  | 0.931    |
> | **ResNet50-sota (Argmax)**            | 0.949        | 0.665       | 0.904   | 0.965  | 0.940    |
> |--------------------------------|--------|-------|-------|-------|-------|
> | **ConvNeXtV2 (Softmax > 0.5)**        | 0.949        | 0.647       | 0.925   | 0.969  | 0.940    |
> | **ConvNeXtV2 (Argmax)**               | 0.979        | 0.748       | 0.956   | 0.984  | 0.944    |
>
> We will add the results to the supplemental material.
>
> **Visualizations for MI and RS samples:**
> We thank the reviewer for the suggestion. We will include visualizations of two representative samples per domain (MI and RS) for each suppression type, analogous to those shown for computer vision.
>
> **Attribution to earlier work for the patch-shuffle idea for analysing shape bias:**
> We thank the reviewer for pointing this out. We fully agree that the use of PatchShuffle for analyzing shape disruptions should be properly attributed. We will update the section on our feature suppression framework and acknowledge their contribution by explicitly citing both [1] and [2] in L147-L149.
>
> **Effect of patch rotation:**
> We thank the reviewer for raising this interesting question. To compare the effects of patch-shuffling versus patch-rotation on shape suppression at different grid sizes, we conducted an analysis on 100 randomly sampled ImageNet images. We computed shape preservation scores using the ESSIM metric for both PatchShuffle and PatchRotation across grid sizes of 3×3, 6×6, 9×9, and 12×12.
>
> | Grid Size | PatchShuffle | PatchRotation |
> |-----------|---------------|----------------|
> | 3×3       | 0.251         | 0.312          |
> | 6×6       | 0.190         | 0.306          |
> | 9×9       | 0.136         | 0.305          |
> | 12×12     | 0.093         | 0.335        |
>
> The reviewer’s hypothesis is partially confirmed: patch rotation alone leads to moderate shape suppression, but with diminishing returns at higher grid sizes. In contrast, PatchShuffle more strongly suppresses shape information across all grid sizes. Interestingly, for very large grids, PatchRotation even shows a slight increase in shape preservation, likely due to re-emergence of global contours, consistent with observations in Figure 6c of the supplemental material.

---

> > ### Comment · Reviewer_mrhF · 2025-08-02
> >
> > Thank you for the thorough rebuttal and the additional experiments. The new results address all of my concerns and fully clarify the questions I raised. In light of this, I am pleased to raise my score to 6 (the highest possible).

---

### Official Review · Reviewer_hYxc · 2025-07-03

**Clarity:** 3
**Significance:** 3
**Originality:** 3
**Rating:** 5
**Confidence:** 3

**Summary:**

This paper challenges the experimental designs and conclusions of Geirhos et al.’s cue-conflict experiment. Instead of doing cue-conflict experiments using style transfer, the paper proposes to use controlled suppression, and measure performance degradation of pre-trained vision models vs. human to understand the bias of models. The paper finds that the CNNs are not inherently biased towards texture, but rather, more biased towards local shape. Additionally, the paper suggests that the domain the models are trained on can also impact the dependence on various features.

**Questions:**

In general, I'm curious to know if the suppression operations can be done by off-the-shelf deep networks, and whether they are better candidates for doing suppressions.
1. Can segmentation models be used for "texture suppression" by replacing the objects with masks, and making the color of each mask the average pixel values of the original region?
2. Can advanced image editing tools be used for perturbing object shapes while keeping textures intact?

**Ethical Concerns:**

["NO or VERY MINOR ethics concerns only"]

**Final Justification:**

I agree with the authors that the suggestions may be more suitable for future work, and I keep my positive score.

**Limitations:**

The limitations are adequately addressed by the authors.

**Quality:**

3

**Strengths And Weaknesses:**

Strength:
1. The idea of the paper is well-motivated, the authors start from the limitations of the Geirhos et al. and propose a new experimental setup to account for the limitations. The paper brings a new perspective that challenges the previously held ideas.
2. The experimental design is convincing, with sufficient details provided.
3. The paper is very well written, it was a pleasant read.

I'm not aware of any pronounced weakness.

---

> ### Author Rebuttal · Authors · 2025-07-29
>
> We are pleased that the motivation and design of our framework were clear and well received. Below, we address the reviewer’s suggestions regarding potential alternative suppression methods.
>
> **On using off-the-shelf networks for suppression (e.g., segmentation + region masking):**
> This is an insightful suggestion. Using pre-trained segmentation networks to identify object regions and replace them with uniform patches (e.g., mean color) could serve as a structured and potentially stronger form of texture suppression that avoids the risk of destructing edges, especially in the context of natural images. This approach would reduce reliance on manually parameterized transformations and may allow for finer control over suppression granularity. However, it depends on reliable segmentation, which may not generalize well across domains such as remote sensing or medical imaging, where segmentation may be less reliable and object structure less clearly defined. On the other hand, many datasets in CV have panoptic segmentation masks (e.g., COCO), which could be also used for a follow up work. We will note it explicitly as a potential extension in future work.
>
> **On using advanced editing tools for shape suppression while preserving texture:**
> We agree that tools from image editing or generation, such as warping or shape-specific inpainting, may offer targeted shape perturbation while avoiding the block-structured artefacts that arise when using Patch Shuffle. Our current framework employs interpretable, model-free transformations chosen to minimize domain-specific artifacts and avoid the complexities introduced by generative or network-based editing tools. However, with ongoing advances in controllable image synthesis, this line of investigation is increasingly viable and we consider it a valuable direction for future refinement of the framework.
>
> **Future work direction:**
> To reflect the reviewer’s suggestions, we will expand the conclusion section to mention segmentation-based and generation-based suppression strategies as promising extensions to increase the precision and applicability of our framework (starting from L320).

---

> > ### Comment · Reviewer_hYxc · 2025-08-01
> >
> > I agree with the authors that the suggestions may be more suitable for future work, and I keep my positive score.

---

### Official Review · Reviewer_CRhF · 2025-07-03

**Clarity:** 3
**Significance:** 3
**Originality:** 3
**Rating:** 5
**Confidence:** 3

**Summary:**

The authors argue that the texture bias results (Geirhos et al.) are based upon the cue-conflict experimental framework with major limitations, e.g. texture feature often containing local shape structures, overloaded texture signals compared to shape signals, biased UI for experimentation. They introduced a suppression-based framework that disentangles texture, shape, and color better, with suppression strength validated by metrics. They then measured the human and model performance while suppressing global/local shape, texture and color and found that CNNs mainly rely on local shape. In addition, the reliance patterns differ across computer vision, medical image and remote sensing domains.

**Questions:**

Questions:
- Please report more statistics on human study that reflects variations in the data, e.g. CI, noise ceiling. 20 subjects are low; compute a post-hoc power estimate.
- Artifacts of block edges when using patch shuffle. Would models be strongly influenced by these artifacts in the images? Please design a control for artifacts

**Ethical Concerns:**

["NO or VERY MINOR ethics concerns only"]

**Final Justification:**

The paper is technically solid. The authors' rebuttal addressed all my concerns. I also believe the work shows an impactful result that directly challenges a widely cited result.

**Quality:**

3

**Strengths And Weaknesses:**

Strength
- Quality
    - Convincing method critique of the cue-conflict experiment (which doesn’t control variables)
    - Nice suppression-based design with clear metrics on validating suppression
    - Extensive experimentation with many model architectures and domains
- Clarity: very well-written
- Originality: the critique and the new framework is novel
- significance: directly challenges a highly-cited claims with a more precise claim of what exactly models rely on

Weaknesses:
- It might be more interesting if you can plot the ROC curve as a function of the hyper parameters for feature suppression - however, I understand that you fixed the hyper parameters in order to make sure the suppression is high for one cue but not the other. It will be great to clarify
- It would be more convincing to train models on cue-suppressed images and test on full-cue images. This would resemble an ablation study

---

> ### Author Rebuttal · Authors · 2025-07-29
>
> We thank the reviewer for their careful reading, constructive critique, and detailed suggestions. We appreciate the recognition of our contributions and are grateful for the opportunity to clarify key aspects of the paper and further strengthen our claims. Below, we address each point raised in detail.
>
> **Plotting the ROC curve as a function of the hyper parameters for feature suppression:**
> We thank the reviewer for raising the point about a more in-depth ablation studying the effects of hyperparameter strength. We were not entirely certain what was meant by the reference to a “ROC curve,” but we considered two possible interpretations. (i) If the comment referred to a more detailed quantitative validation of suppression effects under different suppression strengths, we would like to highlight Figure 8 in the supplemental material, which provides a hyperparameter ablation across several texture suppression techniques using high-frequency energy (HFE) for texture suppression and gradient correlation (GC) for shape preservation. This analysis supported the fixed settings used in the main paper (4.1, Table 2). (ii) Alternatively, if the reviewer referred to how model performance degrades with increasing suppression strength, we note that such an analysis is included in Figure 4, where we vary the suppression strength (e.g., kernel size, σ) for the domain-level comparison experiment. Due to resource constraints, this could not be replicated for the human study, where we selected a single validated suppression strength per feature type. If none of the two interpretations matched the intended concern, we would be happy to provide any additional analysis the reviewer finds relevant.
>
> **Ablation study training on cue-suppressed images and testing on full-cue images:**
> We thank the reviewer for this thought-provoking suggestion. We spent considerable time reflecting on its implications within our team and conducted the requested experiments across all four vision datasets for which quick retraining was feasible (Caltech101, Flowers102, OxfordIIITPet, STL10).
>
> Upon deeper consideration, we realized that this setup (i.e., training exclusively on cue-suppressed images and evaluating on full-cue images) raises a conceptual distinction that is somewhat orthogonal to the main aim of our study. Our framework is designed to investigate what models rely on after training, not what they need during training to generalize. The proposed ablation study targets the latter: removing a feature type during learning and asking whether generalization to full-cue data still occurs. In contrast, our suppression-based evaluation probes which cues are functionally relevant to models that have already converged to a fit on the original distribution. These are distinct questions: learning without access to a particular feature type likely impairs generalization capacity, but this does not necessarily inform which features the model will rely on once generalization is achieved.
>
> Nevertheless, to empirically assess this contrast, we trained ResNet-50 models from scratch following the same protocol as in Experiment 5.2 with the only difference that the training augmentation pipeline consistently applied one of the four suppression transformations (global shape, local shape, texture, color) with a probability of 1.0. This ensured that the respective visual cue was entirely absent during training. We then evaluated the models on full-cue images. All models trained under suppression exhibited substantially reduced performance. This suggests that withholding a specific cue during training hinders the model’s ability to develop robust internal representations that generalize to the full distribution.
>
> | Dataset    | No Global Shape | No Local Shape | No Texture | No Color | All cues |
> |------------|----------------------------|----------------------------|------------------------|----------------------|---------------------|
> | Caltech101 | 0.0851                     | 0.0790                     | 0.0838                 | 0.0524               | 0.7972              |
> | OxfordIIPet | 0.1179                    | 0.1102                    | 0.1089      | 0.1088.          | 0.6214              |
> | Flowers102 | 0.2951                     | 0.2863                     | 0.2657                 | 0.1118               | 0.5186              |
> | STL10      | 0.4300                     | 0.4040                     | 0.3760                 | 0.3860               | 0.7800              |
>
> A technical note: For the color-suppression setting, observed that normalization statistics are tightly tied to the training distribution. We therefore recomputed mean and standard deviation values for grayscale images and used them consistently in both training and testing. The grayscale-trained models were evaluated on grayscale test images to avoid distribution mismatch.
>
> **More statistics to support results: CI, noise ceiling and post-hoc power estimate:**
> We acknowledge that the sample size in the human study (n = 20) is limited, reflecting practical constraints during data collection. We agree that careful statistical treatment is essential in such settings and thank the reviewer for the concrete suggestions.
>
> In response, we have expanded the statistical analysis as follows:
>
> - We computed 95% confidence intervals for human performance, model performance, and their differences (see Table below).
> - We estimated the noise ceiling to assess inter-subject and inter-model consistency. The item-wise human noise ceiling was 0.542 ± 0.055, and the model noise ceiling was 0.646 ± 0.015.
> - We conducted a post-hoc power analysis to enrich the t-test results in the supplemental material (updated Table visualized in this response). Despite the small sample size, the observed effect sizes are large, resulting in maximal statistical power in all conditions.
>
> | Feature Suppression | Human Accuracy (CI)     | ResNet Accuracy (CI)    | Difference (Human - ResNet, CI) |
> |---------------------|-------------------------|--------------------------|----------------------------------|
> | Global Shape        | 0.965 (±0.0067)         | 0.832 (±0.0102)          | 0.133 (±0.0121)                  |
> | Local Shape         | 0.763 (±0.0092)         | 0.276 (±0.0141)          | 0.487 (±0.0168)                  |
> | Texture             | 0.979 (±0.0064)         | 0.795 (±0.0073)          | 0.184 (±0.0097)                  |
> | Color               | 0.999 (±0.0052)         | 0.924 (±0.0035)          | 0.075 (±0.0063)                  |
>
> | Suppression Type       | t-value | p-value | Cohen's d | Power  |
> |------------------------|--------:|--------:|----------:|-------:|
> | Global Shape           |  24.762 |  0.0001 |     12.38 | 1.0000 |
> | Local Shape            |  48.157 |  0.0000 |     24.08 | 1.0000 |
> | Texture                |  79.651 |  0.0000 |     39.83 | 1.0000 |
> | Color                  |  17.382 |  0.0004 |      8.69 | 1.0000 |
>
>
> We will enrich the statistical results in the supplemental material accordingly by including these extended analyses.
>
> **Artifacts of block edges when using patch shuffle and design of control:**
> We agree that disentangling the effects of block-edge artifacts from the intended suppression of local shape continuity is nontrivial. To the best of our knowledge, there is no perfect way to isolate the influence of block structures without also introducing some level of micro distortion to the image’s shape features.
>
> Nevertheless, to probe this question more carefully, we designed a control experiment aimed at partially disentangling these two factors. For grid sizes 2, 4, and 8, we constructed a variant of the PatchShuffle condition in which we superimpose only the 2-pixel-wide block boundaries (1 pixel on either side) onto the original image, preserving the spatial layout of all underlying content. This overlay mimics the grid structure introduced by PatchShuffle but does not shuffle any patches. The images with grid overlay are generated by applying patch shuffle to an image, extracting the 2-pixel wide grid of block structure and pasting it onto a copy of the original image.
>
> We emphasize that even this overlay condition introduces brief shape discontinuities, as the superimposed grid lines may slightly interrupt edge continuity at the boundaries. As such, the overlay condition still jointly reflects two effects: (i) the presence of artificial block edges, and (ii) minor local shape disruption. Nonetheless, the stronger degradation observed under PatchShuffle suggests that major performance drops are not driven solely by block artifacts, but by the combined disruption of local spatial structure.
>
> We will include both the visual examples of this control condition and the corresponding quantitative results in the supplemental material of the revised version. The results for three representative models are shown below:
> | Model            | Grid Size | PatchShuffle Overlay | PatchShuffle |
> |------------------|-----------|-----------------------|--------------|
> | **ResNet50-standard** | 2         | 0.950                 | 0.921        |
> |                  | 4         | 0.809                 | 0.548        |
> |                  | 8         | 0.540                 | 0.069        |
> |        |       |       |           |
> | **ResNet50-sota**     | 2         | 0.983                 | 0.980        |
> |                  | 4         | 0.870                 | 0.837        |
> |                  | 8         | 0.554                 | 0.344        |
> |        |       |       |           |
> | **ConvNeXtV2**        | 2         | 0.991                 | 0.983        |
> |                  | 4         | 0.957                 | 0.859        |
> |                  | 8         | 0.911                 | 0.347        |
>
> We hope this analysis addresses the reviewer’s concerns, and we remain open to further suggestions if additional clarification or controls are desired.

---

> > ### Comment · Reviewer_CRhF · 2025-08-04
> >
> > The authors’ response addressed all my main concerns.
> >
> > Thank you for your careful thoughts on the ablation study. I understand the conceptual distinctions better now. I proposed the ablation to answer the question “does the network need this cue to learn the representation (for generalization)”, for the simplified thought that if a network doesn’t rely on a cue, then removing such cue from training does not hinder its performance on the naturalistic images. However, I can see that the paper emphasizes functional reliance of the model, and that there is distinction between model training and inference.
> >
> > I also appreciate that the authors reported more statistics of the human study. Please include these analyses in the revision.
> >
> > Given the clarifications and new results, I will raise my overall score.

---

### Decision · Program_Chairs · 2025-09-17

**Decision:**

Accept (oral)

**Comment:**

a) The authors challenge the claim that ANN are biased towards textures in the task of image classification. They raise several concerns about the previously used methodology and resolve them to provide a better view of the interaction between shapes and textures in ANNs and in humans.

b) The paper challenges a widely accepted claim.
- the methodology is acknowledged to be better by all reviewers
- extensive exp validation on ANNs and Humans and using mutiliple datasets
- use of appropriate metrics

c) minor / resolved during the rebuttal period

d) The authors revisit an impactful previous work by highlighting drawbacks and limitations and proposing a methodology that resolve these. The results go against the previous claim showing that ANNs are not biased towards textures. The impact of the previous work justifies to put this paper forward with an oral presentation (or at least a spotlight).

e) Initially, all reviewers add a positive opinion about this submission. The rebuttal period resolves most of their concerns improving their opinion about the paper.